# Research on the Preparation of Graphene Quantum Dots/SBS Composite-Modified Asphalt and Its Application Performance

Youfu Lu [1], Nan Shi [2], Mingming Wang [2], Xinyang Wang [3], Liyang Yin [4], Qiang Xu [4] and Pinhui Zhao [4,*]

1   Shandong Hi-Speed Group, Jinan 250098, China; luyoufu1111@163.com
2   Shandong Hi-Speed Construction Management Group Co., Ltd., Jinan 250001, China; 0531shinan@163.com (N.S.); wangmm@lreis.ac.cn (M.W.)
3   Shandong Hi-Speed Testing Engineering Co., Ltd., Jinan 250001, China; wangxinyang1986@sina.com
4   School of Transportation Engineering, Shandong Jianzhu University, Jinan 250101, China; yinliyang01@163.com (L.Y.); longqiang1986@sdjzu.edu.cn (Q.X.)
*   Correspondence: zhaopinhui08@163.com

**Abstract:** This study aims to prepare a graphene quantum dots (GQDs)/styrene-butadiene segmented copolymer composite (GQDs/SBS) as an asphalt modifier using the Pickering emulsion polymerization method. The physicochemical properties of the GQDs/SBS modifier and their effects on asphalt modification were investigated. In addition, the GQDs/SBS modifier was compared with the pure SBS modifier. Research results demonstrated that GQDs could be evenly dispersed into the SBS phase to form a uniform composite. Adding GQDs brings more oxygen-containing functional groups into the GQDs/SBS modifier, thus strengthening the polarity and making it disperse into the asphalt better. Compared with the SBS modifier, the GQDs/SBS modifier presents better thermostability. Moreover, GQDs/SBS composite-modified asphalt achieves better high-temperature performance than SBS-modified asphalt, which is manifested by the increased softening points, complex shear modulus and rutting factors. However, the low-temperature performance decreases, which is manifested by reductions in cone penetration, viscosity and ductility as well as the increased ratio between creep stiffness (S) and creep rate (m), that is, S/m. Furthermore, adding GQDs can improve the high-temperature performance of asphalt mixture, but it influences low-temperature and water stability slightly. GQDs/SBS also have the advantages of simple preparation techniques, low cost and are environmentally friendly. Therefore, they have become a beneficial choice as asphalt cementing material modifiers.

**Keywords:** graphene quantum dots (GQDs); GQDs/SBS composite-modified asphalt; pavement performances; rheology

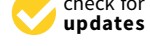



## 1. Introduction

In all pavement forms, asphalt pavement accounts for a very high proportion of road engineering around the world due to its remarkable advantages such as high riding comfort and convenient maintenance. Recently, transportation industries in countries around the world are booming. Increasing traffic loads, especially the growth in heavy-loaded and overloaded vehicles, intensifies damage to original roads. As a result, asphalt pavements may develop different types of early diseases soon after opening to traffic, such as ruts, pavement subsidence, upheaval, etc. [1,2]. These diseases affect the performance and service life of pavements significantly. Nowadays, improving durability and prolonging the service life of asphalt pavement is a key problem that has to be solved in the road field at present. Furthermore, developers often choose superior performance materials for asphalt pavements since excellent pavement structural performances are closely related to material performances. In particular, it is very important to choose a good performance asphalt binder because its quality is directly related to the performance of the asphalt pavement [3]. To obtain ideal performances from asphalt under various climatic and traffic

load conditions, researchers tried adding modifiers such as rubber, resin, polymer and other fillers [4]. Recently, applications of new materials and new technologies achieved great progress. In particular, with the continuous development of nanotechnology, adding nanomaterials into polymer-modified asphalt materials, such as applications of nanoclay, nano-silica, nano-ore and nano-metal [5–7], occurs frequently. This is due to the superior high-temperature and low-temperature stability, durability and water stability of the nanomaterial/polymer composite-modified asphalt. Bhat et al. [8] found that the storage stability and the aging resistance of SBS-modified asphalt binder after adding $Al_2O_3$ were improved significantly. Martínez-Anzures et al. [9] pointed out that the softening point and viscosity of SBS-modified asphalt binder after adding Cloisite 15 A was improved under high temperatures. Golestani et al. [10] investigated the effect of organic montmorillonite (OMMT) in improving the storage stability and physical and rheological performances of SBS-modified asphalt binder. The asphalt mixture prepared by composite materials presented better flexural–tensile strength, elasticity modulus and smaller rutting depth. Bala et al. [11] found that mixing nano-silica could improve the compatibility of polypropylene modified asphalt, and the aging resistance and high-temperature performance of modified asphalt were improved significantly. Alhamali et al. [12] found that the viscosity and ductility of the polymer-modified asphalt binder were increased after nano-silica was added, accompanied by a significant improvement in high-temperature and storage stability. Zhang et al. [13] investigated the influences of nano $TiO_2$, nano-Zn and nano-$CaCO_3$ on the high-temperature and low-temperature performance of SBS and SBR polymer-modified asphalt binders. Results showed that nanomaterials could improve the dispersion of polymers in base asphalt and improve the compatibility between polymers and base asphalt. Therefore, the high-temperature and low-temperature performance of polymer-modified asphalt were improved.

Recently, another type of nanomaterials, that is, carbon nanomaterials, has attracted more and more research attention. Specifically, there are abundant products for the nanomodification of binding materials (referring to asphalt and concrete), including the nanomaterials of the graphene family (e.g., original graphene, monolayer graphene, multi-layer graphene and graphene nanosheets), graphene oxide (GO), single-wall carbon nanotube (CNT) and multi-wall CNTs [14–16]. Based on experimental studies, Liu [17] and Zhu [18] pointed out that GO could promote high-temperature viscoelasticity, strengthen humidity resistance and promote the internal healing ability of asphalt binder. Stratiev [19,20] found that the addition of H-Oil hydrocracked vacuum residual oils (H-Oil VTB) to the straight run vacuum residual oils (SRVRO) increases the oxidation rate of the SRVRO, which leads to a higher rate of asphaltenes formation, and, subsequently, a higher rate of softening point increasing. Santagata [21] stated after a survey that CNT could promote the high-temperature and low-temperature performance of asphalt binders and decrease the influence of the external climate on binder aging. Through a nano-scale study on an atomic force microscope (AFM), Mamun [16] pointed out that the combination of polymer and single-wall CNT composite improved the humidity resistance of the asphalt binder. Furthermore, Goli et al. [22] discussed the influences of CNT as a binder modifier on the performance storage stability of SBS asphalt. They found that the physical, rheological and storage stability of the SBS-modified asphalt binder were improved significantly after adding CNT. In recent years, graphene quantum dots (GQDs) have attracted strong attention for their unique structures, excellent performances and promising application prospects [23–26]. In particular, GQDs, which were prepared by asphaltene, were formed by the connection of abundant carboxyl and hydroxyl functional groups surrounding the 2–5 nm aromatic nucleus. Since GQDs are derived from asphalt, it can be reasonably speculated that GQDs have very good compatibility with asphalt [27]. Compared with other nanomaterials, GQDs contain a lot of carboxyl and hydroxyl on the surface and possess some acidity and surface activity. They are expected to develop some physical or chemical reactions with unsaturated dilute bonds of SBS, thus forming stable chemical crosslinking and developing the synergistic effect between GQDs and SBS effectively.

Nevertheless, carbon nanomaterials must overcome considerable surface tension in order to disperse into asphalt since they have great specific surface area [4]. As a result, the dispersion problem of carbon nanomaterials in SBS-modified asphalt is a key constraint against their development at present. With abundant carboxyl and hydroxyl functional groups on the surface, GQDs show some surface activity and they can be used to prepare Pickering emulsion as nano surfactant [28]. Further, the polymerization of Pickering emulsion is expected to be a new choice to prepare GQDs/SBS composite-modified materials [4,29].

This study aims to discuss the applications of GQDs as an asphalt modifier. For this purpose, the asphalt-based GQDs and SBS were used as the main raw material, and a new GQDs/SBS composite material prepared by the Pickering emulsion polymerization method was used as the modifier of asphalt. A series of chemical analyzers were used to analyze the functional group structures and thermostability of the GQDs and SBS modifier. On this basis, the conventional physical properties and rheological properties of GQDs/SBS composite-modified asphalt and SBS-modified asphalt were compared. Moreover, pavement performances of asphalt mixtures that were prepared using GQDs/SBS composite-modified asphalt and SBS-modified asphalt as binders, were characterized.

## 2. Materials and Methods

### 2.1. Materials

GQDs/SBS modifier was prepared using asphalt-based GQDs and linear SBSas the raw materials. Modified asphalt was prepared using Qinhuangdao AH-70 (PetroChina fuel asphalt Co., Ltd., Qinhuangdao, China)asphalt and GQDs/SBS modifier as raw materials. The conventional performances and four-component compositions of Qinhuangdao AH-70 asphalt are listed in Table 1. SBS, white fluffy rod-like solids, was provided by Yueyang Baling Petrochemical Corp (Sinopec, Yueyang, China). SBS is a linear molecule with an average molecular weight of 100,000 g/mol.

**Table 1.** Conventional Index of Qinhuangdao AH-70 asphalt.

| Index | | Units | Requirement | Results |
|---|---|---|---|---|
| Penetration (25 °C, 5 s, 100 g) | | 0.1 mm | 60–80 | 71 |
| Softening point (R&B) | | °C | ≥45 | 50.2 |
| Ductility (10 °C) | | cm | ≮15 | 35.7 |
| Dynamic viscosity at 60 °C | | Pa·s | ≮160 | 212 |
| Four component | Saturates | % | – | 17.8 |
| | Aromatics | % | – | 42.7 |
| | Resins | % | – | 25.2 |
| | Asphaltenes | % | – | 14.1 |

The deoiling asphalt (DOA, Asphaltenes content is 20%) from SINOPEC Jiujiang company, Jiujiang, China, was used as the raw material, and GQDs with asphaltene polycyclic aromatic hydrocarbon nucleus were prepared by nitric acid oxidation. The specific manufacturing technology was introduced as follows: 10 g DOA powder were added into a 250 mL flask, and 150 mL 65% concentrated nitric acid were added in slowly and stirred continuously. Under the strong stirring, the temperature increased gradually, and the reflux was heated. The reaction lasted for 4 h under 90 °C. After finishing the reaction, it was cooled to room temperature and then diluted with distilled water. It was filtered by a 0.2 um Millipore filter directly rather than neutralized by sodium hydroxide. The residual nitric acids in the filtrate were eliminated through reduced pressure distillation and then dried, thus obtaining nitric acid oxidized GQDs.

## 2.2. Preparation of GQDs/SBS Modifier

The GQDs/SBS modifier was prepared by the Pickering emulsion polymerization method. The preparation process is shown in Figure 1. Firstly, a certain mass of GQDs was dispersed into pure water, and a 5% (mass concentration) GQDs solution was gained through ultrasonic dispersion for 30 min. Meanwhile, SBS particles were dissolved into methylbenzene, and a 20 wt% (mass concentration) SBS methylbenzene solution was prepared. The SBS methylbenzene solution was added to the GQDs solution at a mass ratio of 1:1, and Pickering emulsion was acquired through 5 min high-speed shearing using a BME shearing machine under 4000 r/min. The Pickering emulsion was poured into a clean glass tray with a flat bottom. Finally, the tray was put in a vacuum drying box under 80 °C for 12 h. The Pickering emulsion developed auto polymerization under these conditions, thus obtaining a GQDs/SBS modifier. It can be seen from Figure 2 that the GQDs/SBS modifier is a black solid under room temperature.

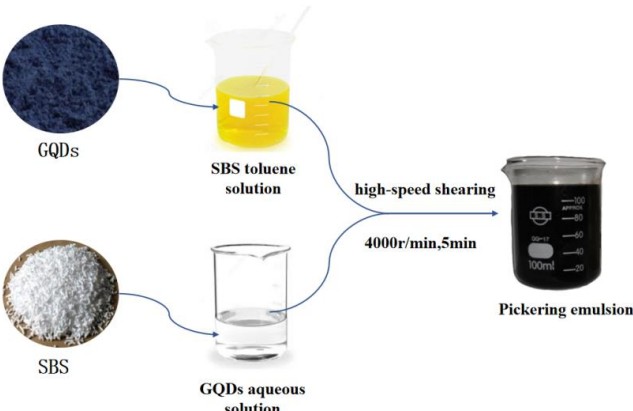

**Figure 1.** The preparation process of GQDs/SBS composite modifier Pickering emulsion.

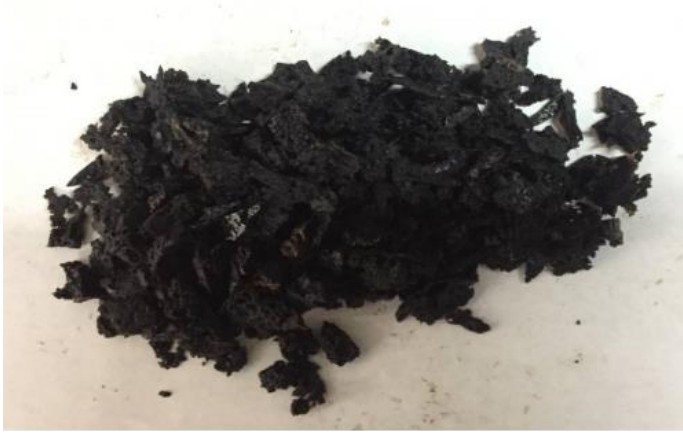

**Figure 2.** GQDs/SBS composite modifier.

## 2.3. Structural Analysis of GQDs/SBS Modifier

### 2.3.1. FT-IR Spectral Analysis

The functional groups and material structures of the SBS modifier and the GQDs/SBS modifier were characterized using a Fourier infrared microscopic analysis spectrometer. Meanwhile, their chemical compositions were analyzed. A Nicolet IS 5-type infrared spectrometer (Thermo Science, Waltham, MA, USA) was used in the experiment. All tests were performed at room temperature. The resolution was 4 cm$^{-1}$, the scanning frequency was 32 times/min and the spectral wavenumber ranged between 4000 and 500 cm$^{-1}$. The samples were prepared by casting a film onto a potassium bromide (KBr) window from a 5% by weight solution in carbon tetrachloride (CCl$_4$).

### 2.3.2. Thermogravimetric Analysis (TGA)

The TGA-100 A (Shanghai all Instrument Equipment Co., Ltd., Shanghai, China) thermal gravimetric analyzer was applied in the experiment for the TGA of the SBS modifier and GQDs/SBS modifier. Under the nitrogen atmosphere, about 7 mg samples were collected and heated from 30 to 600 °C at the constant temperature rising rate of 10 °C/min. The thermostability performances of the two modifiers were evaluated by TG and DTG curves. To assure accuracy and decrease errors, all experiments were performed three times.

### 2.4. Preparation of Modified Asphalt

This study prepared GQDs/SBS composite-modified asphalt and SBS-modified asphalt (control group) using the melting–thawing mixed method. The melted–thawed AH-70 asphalt was collected and then poured into a cylinder container, which was then heated to 180 °C. Subsequently, 3 wt% (asphalt mass) compatilizer (extract oil) and 4 wt% modifier were added successively. Next, the mixture was processed by high-speed shearing for 30 min at the rate of 4000 r/min. Later, the temperature was lowered to 170 °C, and the stirring rate was 750 r/min. In total,0.25 wt% stabilizer was added and stirred continuously for 3 h. After full development, modified asphalt with stable performance was obtained.

### 2.5. Performance Characterization of Modified Asphalt
### 2.5.1. Characterization of Physical Properties of Modified Asphalt

With reference to Test Regulations on Highway Engineering and Asphalt Mixture (JTG E 20-2011), various physical properties of modified asphalt, including cone penetration, softening point, ductility and viscosity were characterized.

### 2.5.2. Rheological Test

The rheological properties of modified asphalt samples were characterized using the dynamic shear rheometer (DSR, TA, New Castle, DE, USA). The clamp chose the parallel plates with diameters of 8 mm and 25 mm, respectively. Firstly, the linear viscoelastic interval of the samples was determined through a stress and strain scan. Secondly, a small-angle vibration shearing test was carried out within the determined linear viscoelastic interval. The scanning results of isothermal frequencies (0.1–50 mads) were acquired under 30, 45, 60, and 75 °C. The specific operation process was introduced as follows: first, put about 0.1 g of the samples on the lower plate of the parallel plates. Second, install the parallel plates on the rheometer, and set the initial temperature. After the samples are softened, lower the upper plate to squeeze some samples. Finally, set the interval between the parallel plates to 1 mm (25 mm plate) or 1.5 mm (8 mm plate). The temperature scanning ranged from 58 to 95 °C. The temperature rising rate was 1 °C/min, and the frequency was l0 rad/s. The multi-stress repetitive creeping test was carried out under 100 and 3200 Pa. Each stress cycle number was set to 10, and each circle had 1 s stress loading and 9 s relaxation.

The bending dye rheometer (BBR, ATS, Butler, PA, USA) was used to measure the creep properties of the asphalt under low temperatures. The combination of BBR and DSR can present relatively comprehensive rheological information on asphalt under the used temperature. BBR uses the small beam principle in engineering to characterize the cracking trend of the asphalt upon temperature drop, through which two indexes could be gained: creep stiffness (S) and variation rate of stiffness with time (m). To avoid the cracking phenomenon of the asphalt under low temperatures, Peformance Grade (PG) classification norms require that the S for 60 s loading of BBR should be no higher than 300 MPa and the m value should be no smaller than 0.3. The temperature of the BBR test ranged between −18 and −24 °C.

In this study, the viscoelasticity within a wide-frequency and wide-temperature range was gained by the time–temperature equivalence principle. Such viscoelasticity with a very large span in orders of magnitude can hardly be measured directly. The time–temperature equivalence principle elaborates that influences of extended time (or decreased frequency)

on mechanical properties of materials are equivalent to temperature rise. Under conditions meeting the time–temperature equivalence principle, various viscoelastic parameters measured by experiment can be used to synthesize curves using translocation factors.

*2.6. Performance Characterization of Asphalt Mixture*

The SBS-modified asphalt (control group) and the prepared GQDs/SBS composite-modified asphalt were used as binders, respectively. The AC-20 asphalt mixture, which is commonly used in asphalt pavement surfaces, was chosen to design asphalt mixture by the Marshall Design method according to China's Construction Technological Norms on Highway Asphalt Pavement (JTG F40-2004). The grading curve is shown in Figure 3. Combining with engineering experiences, the optimal oil–stone ratio was determined to be 4.5 based on the target voidage of 4.0%. In this study, all evaluated asphalt mixtures had the same grading and optimal asphalt content.

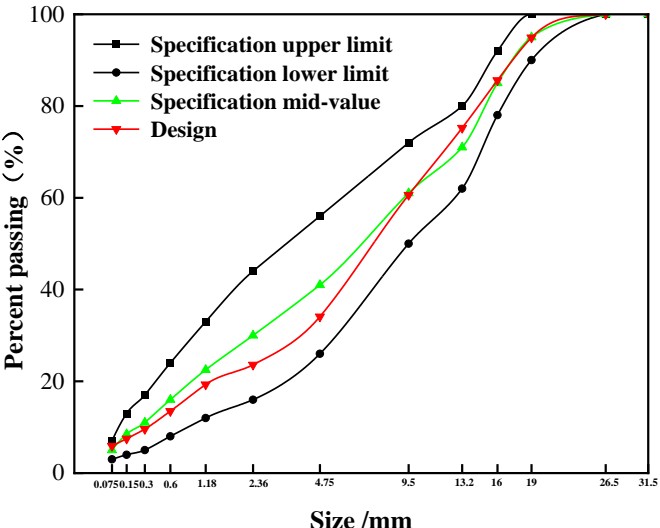

**Figure 3.** AC20 aggregate gradation curves.

Two asphalt mixtures were molded into specimens according to Test Regulations on Highway Engineering and Asphalt Mixture (JTG E 20-2011) of China. The properties of the asphalt mixtures, including high-temperature performance, low-temperature performance and water stability, were analyzed. Since the grading and asphalt consumptions of the two asphalt mixtures were the same, the volume indexes were similar, and their differences in performance indexes were mainly determined by the different performances of the asphalt cements.

## 3. Results and Discussions
### 3.1. Chemical Properties of Modifiers
#### 3.1.1. FTIR Functional Group Analysis

The FT-IR spectra of the SBS modifier and GQDs/SBS composite modifier are shown in Figure 4. The IR region (wavenumber from 4000 to 400 cm$^{-1}$) was divided into a functional group zone (wavenumber from 4000 to 1330 cm$^{-1}$) and fingerprint zone (wavenumber from 1330 to 400 cm$^{-1}$) [30]. It can be seen from Figure 4 that SBS shows obvious methylene C-H asymmetric and symmetric stretching vibration peaks at 2917 and 2848 cm$^{-1}$ as well as multiple absorption peaks between 3100–2950 cm$^{-1}$, which were stretching vibration absorption peaks of unsaturated hydrocarbons. The absorption peaks occurring simultaneously at 1630, 1600, 1560, and 1422 cm$^{-1}$ corresponded to the stretching vibration of the aromatic ring skeleton (–CH$_2$–). The vibration within 1390~1000 cm$^{-1}$ is the stretching vibration of the –C–O bond and single-bond skeleton vibration of C–C. In addition, absorption peaks near 697, 730, and 749 cm$^{-1}$ were caused by the vibration absorption of

single substituted benzene. The absorption peak near 972 cm$^{-1}$ was caused by the twisting vibration of the C=C bond, while the absorption peak near 915 cm$^{-1}$ is the infrared characteristic absorption peak of polybutadiene caused by out-of-plan swinging and vibration of =CH$_2$. As the petroleum asphalt-based GQDs are added, the peaks of the GQDs/SBS composite modifiers at these points are all strengthened. Moreover, a wide adsorption peak occurs at 3307 cm$^{-1}$, which is a combined peak of hydroxyl and amidogen stretching vibrations of petroleum asphalt-based GQDs. Meanwhile, there are obvious shoulder peaks at 1650–1580 cm$^{-1}$, which are stretching vibration peaks of the benzene ring. This reflects that GQDs and SBS develop polymerization reactions to form stable covalent bonds. These covalent bonds are enough to avoid phase separation which might occur during the simple physical mixing manufacturing of nanocomposites. In addition, modified asphalt contains more oxygen-containing functional groups (e.g., –C=O and –C–O). Furthermore, these functional groups mainly come from GQDs, and the increased oxygen content can also improve the polarity of GQDs/SBS composite modifiers, thus increasing their compatibility with asphalt [31].

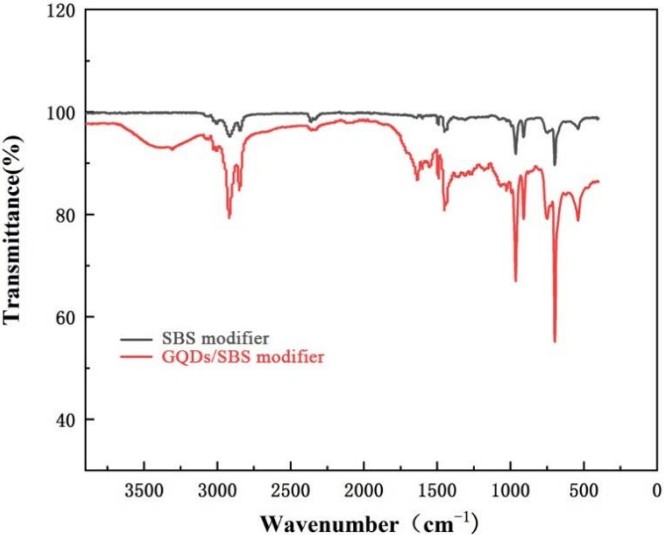

**Figure 4.** FTIR spectrum of GQDs/SBS composite modifier and SBS modifier.

3.1.2. TGA

The thermostability of the modifier is an important property that has to be considered when analyzing the structural characteristics of asphalt binders. In this study, the thermal stability of GQDs/SBS composite modifiers and the SBS modifier were discussed by TGA. It can be seen from Figure 5 and Table 2 that the TGA curves of the GQDs/SBS composite modifier and SBS modifier present the same trend, and they both experienced two major stages of mass loss. However, the thermodynamic behaviors of the GQDs/SBS composite modifier and SBS modifier are significantly different. The GQDs/SBS composite modifier and SBS modifier both enter the first stage of mass loss before 340 °C. In this stage, mass loss is mainly attributed to volatilization of crystal water adsorbed onto the sample surface as well as decomposition of some oxygen-containing functional groups in molecules (–OH and –COOH). Since the GQDs surface has a lot of oxygen-containing functional groups, the mass loss rate of the GQDs/SBS composite modifier is far higher than that of the SBS modifier in the first stage. The second stage of mass loss occurs in the temperature range of 340~490 °C. The mass loss of modifiers is mainly attributed to the decomposition of SBS into small molecules and volatilization. This is the major stage of mass loss. It can be seen from the TG curve that the initial decomposition temperatures of the GQDs/SBS composite modifier and SBS modifier are at about 416 °C (the tangent initial point of TG is about 416 °C), and the pyrolysis termination temperature is about 480 °C. The pyrolysis termination temperature of the GQDs/SBS composite modifier (478.8 °C) is slightly lower than that of the SBS

modifier (479.9 °C), showing a very small difference. After finishing the pyrolysis, the residual mass of the GQDs/SBS composite modifier is 1.78%, and the mass change is 98.22%. The residual mass of the SBS modifier is only 0.05%, and the mass changes are 99.95%. This demonstrates that within this temperature range, the SBS modifier almost loses weight completely under the $N_2$ atmosphere, and it is decomposed into small molecules and then volatilized without producing any residual carbons. However, the GQDs/SBS composite modifier is decomposed incompletely under an $N_2$ atmosphere, and it will produce some residual carbons. This is because the carbon nucleus is the major structural unit of GQDs in the GQDs/SBS composite modifier. After surface oxygen-containing functional groups are lost in the first stage of mass loss, the residual carbon nucleus has very good thermostability, and it will not be decomposed again, thus resulting in a high residual mass. The maximum mass loss rate points (DTG peak value) of the GQDs/SBS composite modifier and SBS modifier are at 460 °C. The mass-loss rate of the SBS modifier (17.08%/min) is 0.85%/min higher than that of the GQDs/SBS composite modifier (16.23%/min). To sum up, the GQDs/SBS composite modifier has better thermostability than the SBS modifier. In other words, adding GQDs improves the thermostability of the SBS modifier.

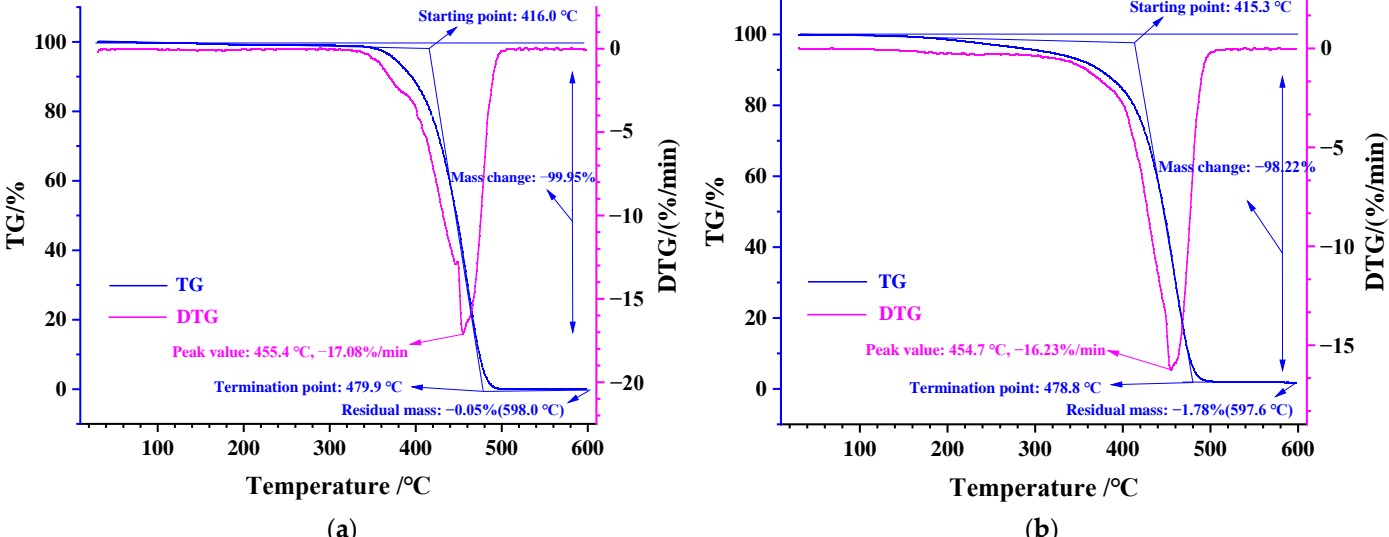

**Figure 5.** Thermal stability analysis of (**a**) SBS modifier and (**b**) GQDs/SBS composite modifier.

**Table 2.** TG and DTG parameters of SBS modifier and GQDs/SBS composite modifier.

| Modifier Type | TG | | | DTG | |
|---|---|---|---|---|---|
| | Starting Temperature/°C | Termination Temperature/°C | Residual Mass/% | Peak Temperature/°C | Pyrolysis Rate/%/min |
| SBS | 416.0 | 479.9 | 0.05 | 455.4 | −17.08 |
| GQDs/SBS | 415.3 | 478.8 | 1.78 | 454.7 | −16.23 |

### 3.2. Conventional Physical Properties of Modified Asphalt

The physical properties of the GQDs/SBS composite-modified asphalt and SBS-modified asphalt are listed in Table 3. Clearly, the cone penetration and ductility of the GQDs/SBS composite-modified asphalt decrease, while the softening point increases compared with those of the SBS-modified asphalt. This implies that adding GQDs can improve the high-temperature performance of SBS-modified asphalt but decrease the low-temperature performance to some extent.

**Table 3.** Physical property of SBS-modified asphalt and GQDs/SBS composite-modified asphalt.

| Index | SBS | GQDs/SBS | Test Method |
|---|---|---|---|
| Softening point/°C | 65.6 | 71.1 | T0606 |
| Penetration/0.1 mm | 51.6 | 47.9 | T0604 |
| Ductility(5 °C)/cm | 24.8 | 24.5 | T0605 |

In addition, temperature influences the high-temperature flowing characteristics of asphalt. Flow characteristics of different samples show different degrees of sensitivity to temperature changes. In other words, asphalts have different temperature sensitivities. There are high-temperature zones and low-temperature zones. The temperature sensitivity of the high-temperature zone is closely related to the construction of asphalt mixture, the pumping of asphalt and other construction characteristics. In this study, a Brookfield rotary viscosimeter was applied, and the 27 # rotors were applied to the viscosity of the GQDs/SBS composite-modified asphalt and SBS-modified asphalt within the temperature range of 110–175 °C. The variation curves of viscosity with temperature are shown in Figure 6a. With the increase in temperature, the viscosity values of both the GQDs/SBS composite-modified asphalt and SBS-modified asphalt decline sharply in the beginning and then become stable. This is because modified asphalt changes from a non-Newtonian body to a Newtonian body gradually under high temperatures. Given the same temperature, the viscosity of GQDs/SBS composite-modified asphalt is lower than SBS-modified asphalt. With the increase in temperature, differences between the GQDs/SBS composite-modified asphalt and SBS-modified asphalt decrease. This reflects that chemical crosslinking between GQDs and SBS is disadvantageous to the strength of polymers.

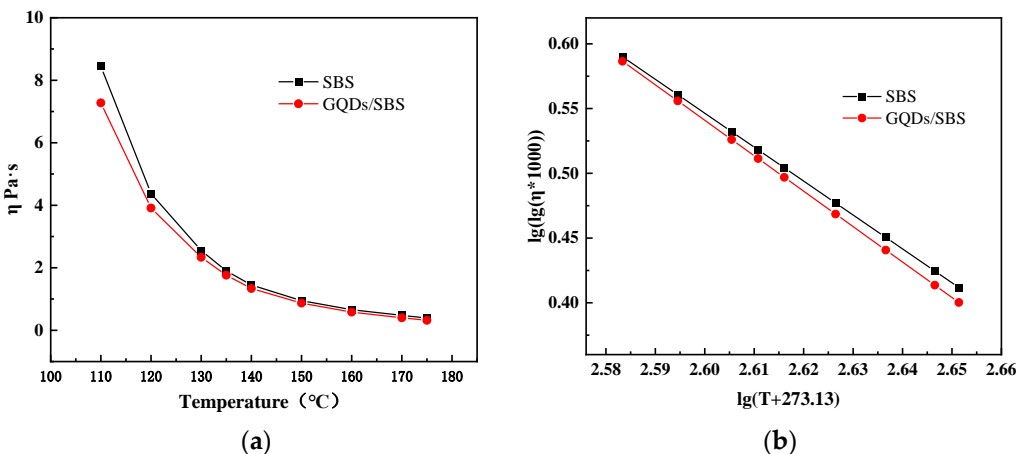

**Figure 6.** Viscosity–temperature performance of SBS and GQDs/SBS composite-modified asphalt. (**a**) Viscosity -temperature curve; (**b**)Viscosity temperature curve after fitting with Saal formula.

The Saal model (Equation (1)) proposed by ASTM D2493 was further applied to process the viscosity–temperature curves. It can characterize the temperature sensitivities of the GQDs/SBS composite-modified asphalt and SBS-modified asphalt.

$$\lg(\lg\eta * 1000) = n + m \cdot \lg (T + 273.13) \tag{1}$$

where $m$ refers to the slope of the regression line; $n$ denotes the intercept of the regression line on the $\lg(\lg\eta*1000)$ axis; $\eta$ is the viscosity (Pa·s); $T$ is the temperature (°C). Saal fitting curves of the viscosity–temperature curves of the GQDs/SBS composite-modified asphalt and SBS-modified asphalt are shown in Figure 6b. The parameters of the corresponding Saal model are listed in Table 4. Moreover, m in the Saal model was defined as viscosity–temperature sensitivity (VTS). The smaller absolute value of VTS indicates that viscosity changes more slowly with temperature, and the temperature sensitivity is better.

**Table 4.** Viscosity–temperature curve fitting equation of SBS and GQDs/SBS composite-modified asphalt.

| Asphalt Type | Viscosity–Temperature Curve Fitting Equation | VTS | $R^2$ |
|---|---|---|---|
| SBS | lg(lg$\eta$*1000) = −2.6184 * (T + 273.13) + 7.3543 | −2.6184 | 0.99732 |
| GQDs/SBS | lg(lg$\eta$*1000) = −2.7364 * (T + 273.13) + 7.6555 | −2.7364 | 0.99959 |

It can be seen from Figure 6 and Table 4 that the absolute value of VTS of the GQDs/SBS composite-modified asphalt is higher than that of the SBS-modified asphalt, indicating that adding GQDs is disadvantageous for the temperature sensitivity of SBS-modified asphalt.

### 3.3. Rheological Properties of Modified Asphalt

The usability of asphalt pavement is determined, to a very large extent, by the viscoelastic properties of the modified asphalt binder. The linear viscoelasticity of modified asphalt is very sensitive to the motion and interaction of polymer molecular chains. Moreover, the complexity of different high-molecular polymer modification systems may influence the internal structure of modified asphalt, thus influencing the rheological characteristics of asphalt. Rheological parameters in the linear viscoelasticity interval are independent of changes regarding stress and strain, and they are only related to the properties of the materials [32]. Therefore, linear viscoelasticity and dynamic rheological tests are very effective methods to elaborate on the influences of modifiers on the performances of modified asphalt and to study the influences of polymers on the viscoelasticity of asphalt.

#### 3.3.1. Frequency Scanning under Middle and High Temperature

The major curve of the GQDs/SBS composite-modified asphalt and SBS-modified asphalt at 30 °C is shown in Figure 7. This curve was obtained from translocations of frequency scanning curves at 30, 45, 60, and 75 °C. It can be seen from Figure 7 that within the whole frequency scanning range, given the same frequency, the modulus of a complex number (G*) of the GQDs/SBS composite-modified asphalt is higher than that of the SBS-modified asphalt. Moreover, the major curves of the GQDs/SBS composite-modified asphalt and SBS-modified asphalt differ significantly in the low-ω zone. However, such difference decreases with the increase of frequency. According to the time–temperature equivalence principle, the low-frequency zone corresponds to the high-temperature zone. Hence, the GQDs/SBS composite-modified asphalt has a better high-temperature performance than SBS-modified asphalt. In other words, adding GQDs improves the rutting resistance of the SBS-modified asphalt. Additionally, the G* values of the GQDs/SBS and SBS-modified asphalt in the high-ω zone are close to the same value, indicating that GQDs influence the viscoelasticity performance of the SBS-modified asphalt in the high-ω zone.

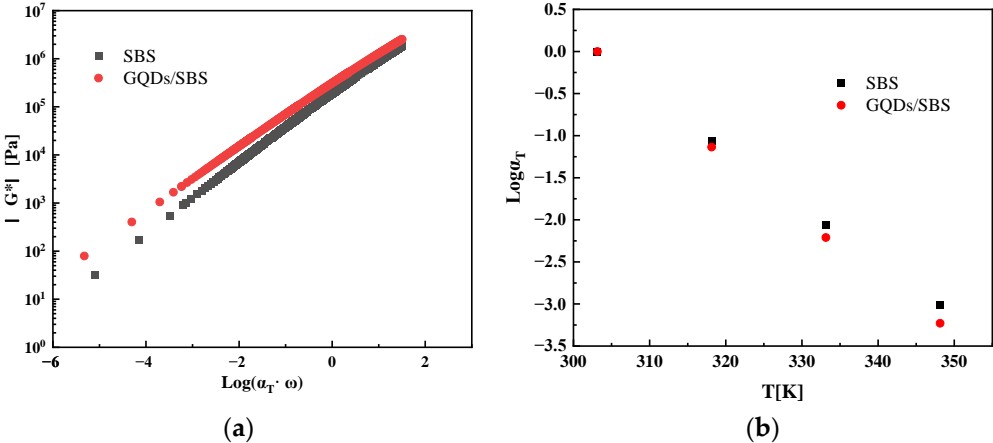

**Figure 7.** (**a**) Master curves for GQDs/SBS composite-modified asphalt and SBS-modified asphalt at the reference temperature of 30 °C and (**b**) Variations of the translocation factor with temperature.

It can be seen from major curves in Figure 7a that the time–temperature equivalence principle is highly applicable to GQDs/SBS composite-modified asphalt and SBS-modified asphalt. Variations of the translocation factor with temperature are shown in Figure 7b. Obviously, the translocation factors of GQDs/SBS composite-modified asphalt and SBS-modified asphalt are significantly different. The variations of translocation factor with temperature were fit using the Arrhenius-like equation (Figure 8). Differences in the translocation factors of different samples can be distinguished quantitatively by the activation energy of the Arrhenius-like equation. The activation energy is related to the temperature sensitivity of materials. This further proves that adding GQDs improves the high-temperature performance of the SBS-modified asphalt.

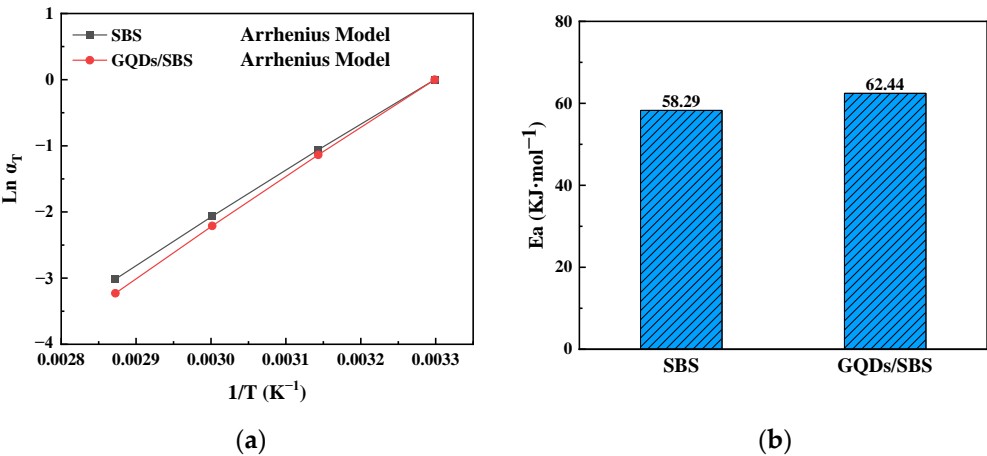

| (a) | (b) |

**Figure 8.** (**a**) Shifting factor versus temperature and (**b**) activation energy for GQDs/SBS composite-modified asphalt and SBS-modified asphalt.

### 3.3.2. Temperature Scanning

In this study, temperatures of asphalt samples within a wide range (58–95 °C) were scanned. The variations of storage modulus (G′) and loss modulus (G″) with temperature are shown in Figure 9. Both G′ and G″ decrease dramatically with the increase in temperature. The reduction rates of the GQDs/SBS composite-modified asphalt and SBS-modified asphalt are different and finally, tend to be stable. Within a wide temperature range, the reduction rates of the G′ and G″ of the GQDs/SBS composite-modified asphalt with the temperature rise are lower than those of SBS-modified asphalt. This reflects that compared to SBS-modified asphalt, the GQDs/SBS composite-modified asphalt has better temperature sensitivity within a wide range.

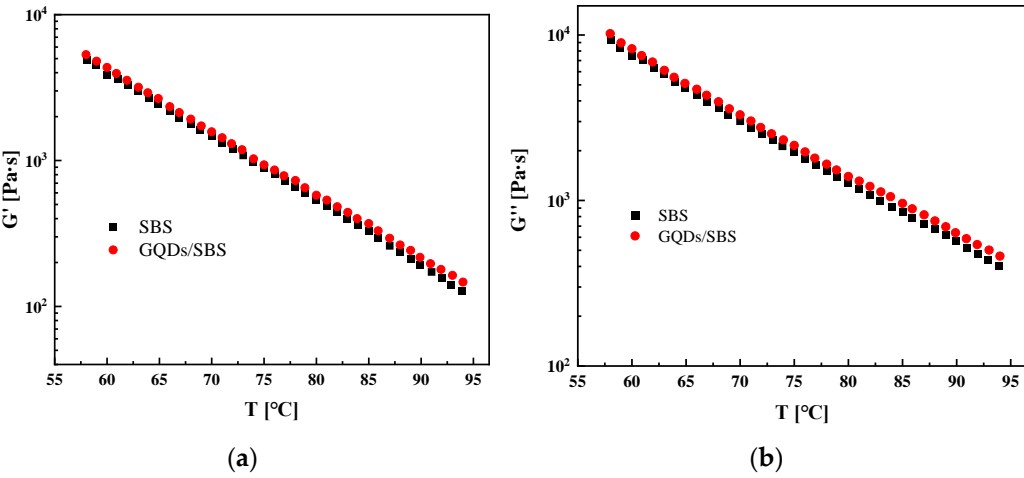

| (a) | (b) |

**Figure 9.** (**a**) Variation of storage modulus G′ with temperature and (**b**) Variation of loss modulus G″ with temperature.

The rutting resistance of asphalt can be characterized by the rutting factor G*/sinδ and the failure temperature which is gained when G*/sinδ = 1.0 kPa. The higher the G*/sinδ and failure temperature, the better the high-temperature stability of asphalt. It can be seen from Figure 10 that in the middle-temperature and high-temperature intervals, asphalt is mainly in the sticky flow state. The elasticity and strength of the system are provided by polymers. The variation laws of the G*/sinδ of the GQDs/SBS composite-modified asphalt and SBS-modified asphalt with temperature are consistent with the variation laws of G′ and G″. Their temperatures at G*/sinδ = 1.0 Kpa are 84.82 and 86.20 °C, respectively. This further demonstrates that GQDs bring higher hardness of SBS-modified asphalt so that SBS-modified asphalt presents better mechanical properties and better resistance to deformation.

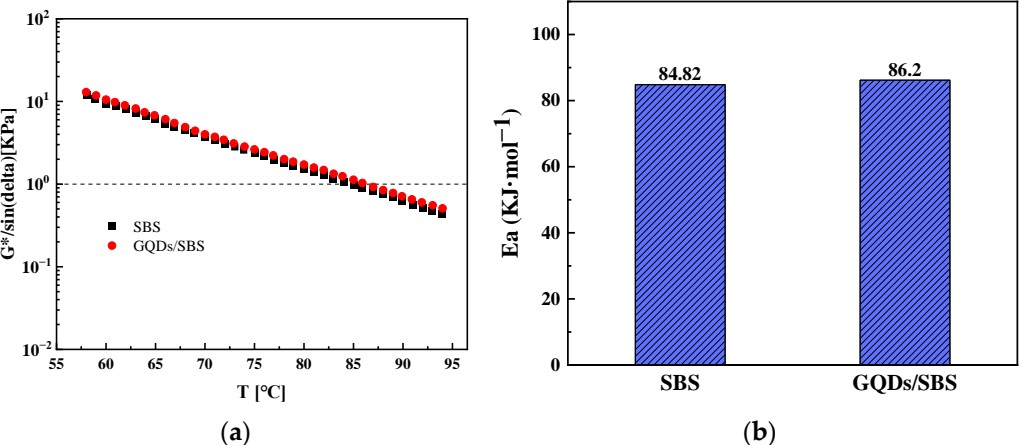

**Figure 10.** (**a**) Evolutions with temperature of the G*/sinδ and (**b**) Temperature calculated at the point of G*/sinδ = 1.0 KPa for SBS-modified asphalt and GQDs/SBS composite-modified asphalt.

### 3.3.3. MSCR

After the reciprocal action of vehicle loads for a long period, asphalt pavement may develop shear creep deformation and form ruts. A multi-stress cyclic creep (MSCR) test is an index used to evaluate the high-temperature performance of modified asphalt in recent years. MSCR usually provides 10 loading cycles to samples. In each cycle, loads are applied for 1s, and then the stress is eliminated for resilience for 9 s. In this study, MSCR tests were carried out at 60 °C under two stress levels (100 Pa and 3200 Pa). In MSCR tests, the recovery rate (R) and unrecoverable compliance (Jnr) could be calculated from the recoverable and unrecoverable strains, respectively.

The strain responses of the GQDs/SBS composite-modified asphalt and SBS-modified asphalt after 10 cycles at 60 °C and two stress levels (100 Pa and 3200 Pa) are shown in Figure 11. For one creep–recovery cycle, the strain of the GQDs/SBS composite-modified asphalt at the end of the creep stage and its strain at the end of the recovery stage are smaller than those of SBS-modified asphalt, which are attributed to the added GQDs.

For the quantitative comparison of high-temperature performance between the GQDs/SBS composite-modified asphalt and SBS-modified asphalt, the parameters of R and Jnr of the two samples at 60 °C under two stress levels (100 Pa and 3200 Pa) are shown in Figure 12. With the addition of GQDs, R increases while Jnr decreases. Given the same conditions, the R of the GQDs/SBS composite-modified asphalt is higher than that of the SBS-modified asphalt, while Jnr is smaller. This implies that adding GQDs increases the high-temperature rutting resistance of asphalt. In addition, the R values of both the SBS-modified asphalt and GQDs/GQDs composite-modified asphalt decrease with the increase of stress. Meanwhile, the Jnr of the two samples increases to some extent. In a word, increasing vehicle loads may weaken the recoverable capacity of asphalt pavement significantly under high temperatures in summer, thus causing rutting damages.

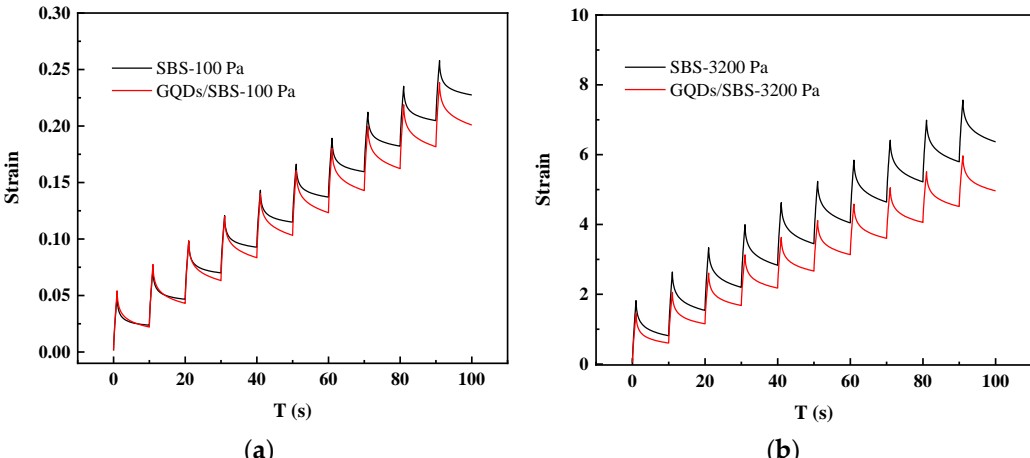

**Figure 11.** Strain response of SBS-modified asphalt and GQDs/SBS composite-modified asphalt (**a**) in 10 cycles at 60 °C 100 Pa and (**b**) in 10 cycles at 60 °C 3200 Pa.

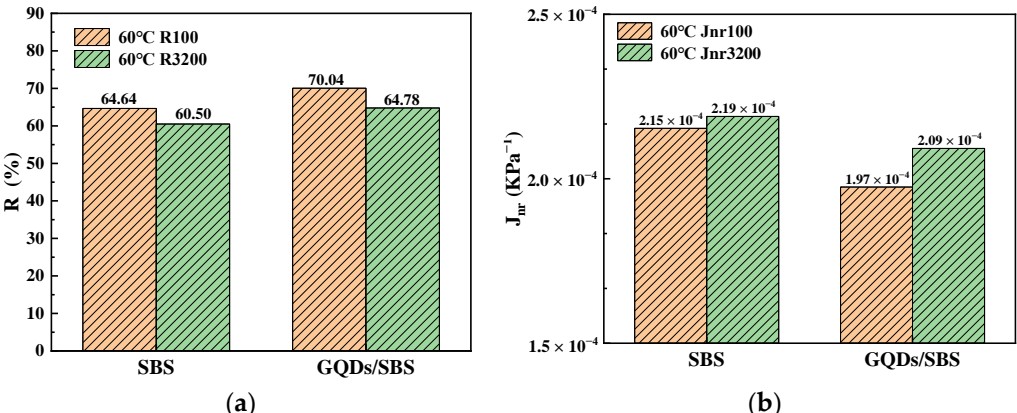

**Figure 12.** (**a**) The percent recovery and (**b**) Non-recoverable creep compliance calculated from MSCR test at 60 °C and two stress levels for SBS-modified asphalt and GQDs/SBS composite-modified asphalt.

### 3.3.4. Low-Temperature Creep Properties

Asphalt pavement may crack under low temperatures. Since there is a binding force between the asphalt mixture layer and the lower layer, it will hinder shrinkage and produce translocations, thus generating tensile stress. Cracks occur when the tensile stress exceeds the tensile strength of the asphalt mixture. This requires the asphalt to have a high creep rate to release stresses generated under low temperatures or in the cooling process.

Since DSR cannot test asphalt which has considerable hardness under low temperatures, BBR is usually applied to measure the creep properties of asphalt when the temperature is very low. BBR uses the small beam principle to characterize the cracking trend of asphalt when temperature declines. Tow indexes can be gained from BBR: creep stiffness (S) and creep rate (m). These two indexes are used to characterize the load resistance and relaxation ability of asphalt. If S is too large, the possibility of cracking is high. If m is relatively low, the relaxation ability is insufficient to release stress produced by the reduction of temperature and the probability of cracking increases.

Variations of the S and m of the GQDs/SBS composite-modified asphalt and SBS-modified asphalt at −18 and −24 °C, which are measured by BBR with time, are shown in Figure 13. S drops quickly with the increase of loading time, while m increases significantly. The variable rates of S and m are different. Given the same loading time, the m of the GQDs/SBS composite-modified asphalt at −18 °C is smaller than that of SBS-modified asphalt. However, the m of the GQDs/SBS composite-modified asphalt at −24 °C is higher. S presents the opposite variation trend. To further compare the low-temperature

performance of the GQDs/SBS composite-modified asphalt and SBS-modified asphalt, S/m at 60 s was used to characterize the low-temperature crack resistance of asphalt. The lower S/m indicates the stronger crack resistance of asphalt and better low-temperature performance. It can be seen from Figure 14 that given the same temperature, the S/m of the GQDs/SBS composite-modified asphalt is higher than that of the SBS-modified asphalt, indicating that the GQDs/SBS composite-modified asphalt has poor low-temperature crack resistance.

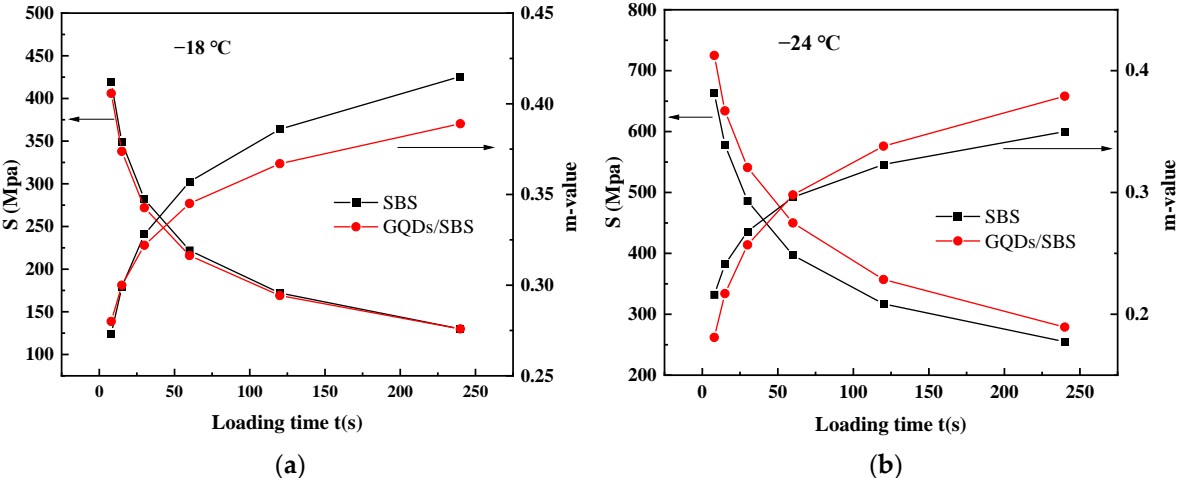

**Figure 13.** Evolution of S and m-value versus loading time for GQDs/SBS composite-modified asphalt and SBS-modified asphalt at (**a**) −18 and (**b**) −24 °C.

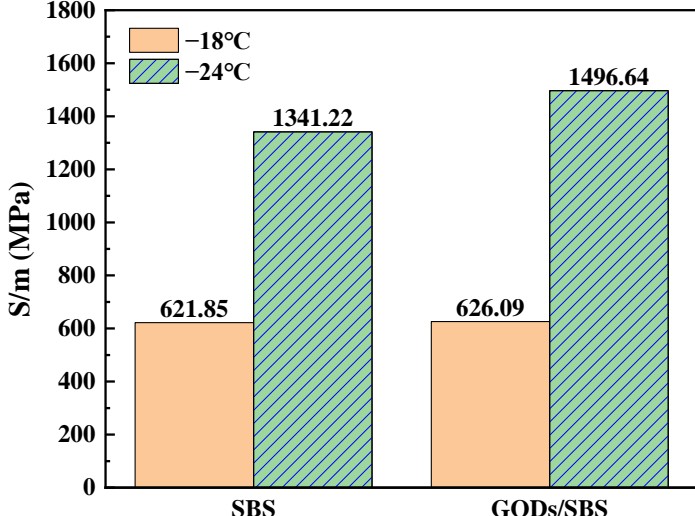

**Figure 14.** The ratio of creep stiffness S and m of SBS-modified asphalt and GQDs/SBS composite-modified asphalt.

## 4. Pavement Performance Test Analysis of Mixture

### 4.1. High-Temperature Stability

In the present study, the high-temperature stabilities of the SBS-modified asphalt mixture and the GQDs/SBS composite-modified asphalt mixture were evaluated by the dynamic stability in the high-temperature (60 °C) rutting test. The high-temperature stability test results are shown in Figure 15.

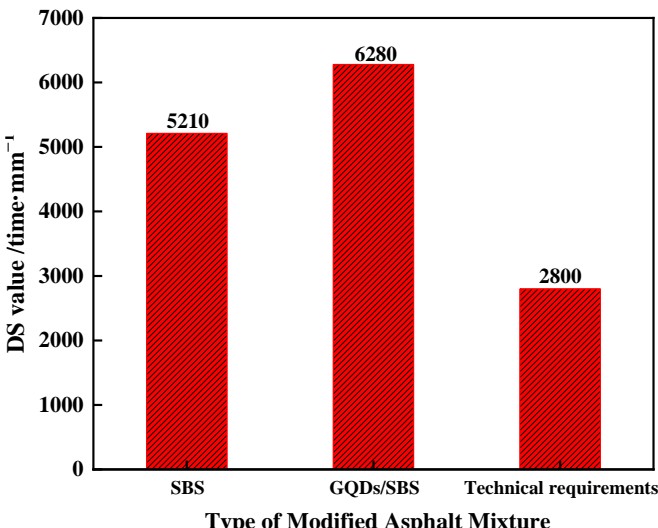

**Figure 15.** Results of wheel tracking test.

It can be seen from Figure 15 that the dynamic stability of the GQDs/SBS composite-modified asphalt mixture is significantly higher than that of the SBS-modified asphalt mixture, indicating that adding GQDs increases the cohesive force of asphalt. Moreover, more compact structures are formed by adjusting the skeleton of asphalt mixtures in the compacting process, thus increasing the internal friction angle. With the increase of cohesive force and internal friction angle, the shear strength of the asphalt mixture increases, thus making it equipped with good high-temperature stability.

*4.2. Low-Temperature Crack Resistance*

The resistance of the asphalt mixture to low-temperature cracking performance was evaluated through a low-temperature small beam bending test. Small beam specimens (250 mm (Length)*30 mm (Width)*35 mm (Height)) were used in the test, and the loading rate and temperature were set to 50 mm/min and −10 °C, respectively. The low-temperature crack resistance test results are shown in Figure 16.

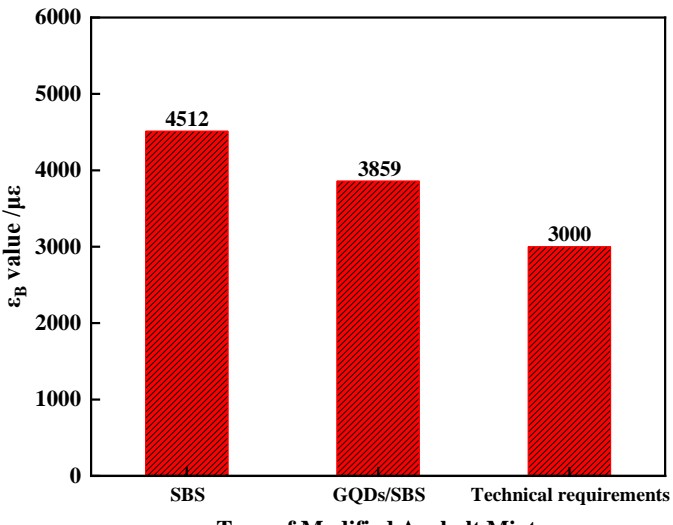

**Figure 16.** Results of beam bending test.

It can be seen from Figure 16 that the maximum bending strain of the GQDs/SBS composite-modified asphalt mixture at low-temperature failures is 14.5% lower than that of the SBS-modified asphalt mixture. However, all test results meet the standard requirements,

indicating that adding GQDs decreases the tenacity and temperature sensitivity of the asphalt mixture under a low-temperature state. As a result, the low-temperature crack resistance declines accordingly.

### 4.3. Water Stability

The water stabilities of two modified asphalt mixtures were evaluated by the freeze–thaw splitting test. After freeze–thaw cycles of specimens based on the Marshall test, freeze–thaw splitting residual strength was tested, thus enabling us to analyze the resistance of the asphalt to water damage under tough environments. The water stability test results are shown in Figure 17.

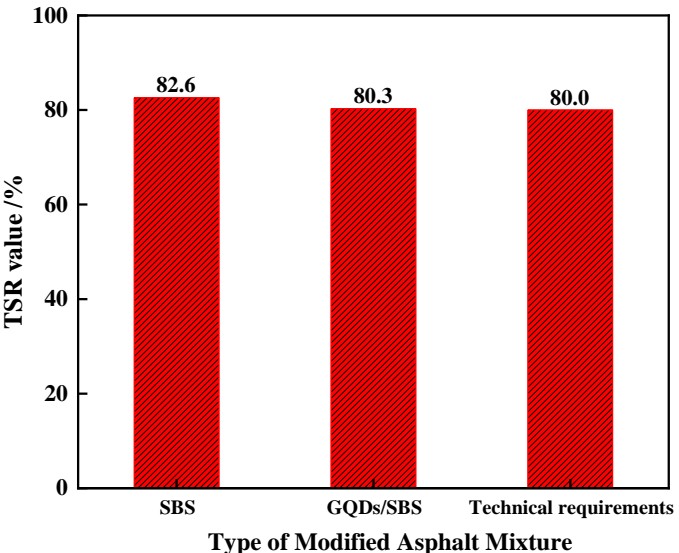

**Figure 17.** Results of immersion Marshall test.

It can be seen from Figure 17 that the residual strengths of the GQDs/SBS composite-modified asphalt mixture before and after freezing and thawing decrease compared with those of the SBS-modified asphalt mixture. This reveals that adding GQDs decreases the adhesion between asphalt and aggregate, thus decreasing the resistance of the asphalt mixture to water damage. However, the residual strength meets the requirements of technical specifications. This implies that the prepared GQDs/SBS composite modifier influences the water stability of the mixture slightly.

### 5. Conclusions

In this study, the GQDs/SBS composite modifier was prepared using the Pickering emulsion polymerization method. In addition, the physical and chemical properties of the GQDs/SBS composite modifier, physical and rheological properties of the binders, as well as pavement performances of the GQDs/SBS composite-modified asphalt mixture were investigated. According to results and discussions, some conclusions could be drawn:

(1) The GQDs/SBS composite modifier is prepared by the simple Pickering emulsion polymerization method. GQDs can evenly disperse into the SBS modifier to form a uniform composite. The GQDs/SBS composite modifier contains more oxygen-containing functional groups than the SBS modifier. Furthermore, the pyrolysis rate of the GQDs/SBS composite modifier is lower than the SBS modifier, and its residual mass is higher, thus showing better thermostability.

(2) The conventional physical properties and rheological properties of the GQDs/SBS composite-modified asphalt and SBS-modified asphalt are compared. The GQDs/SBS composite-modified asphalt shows a higher softening point, complex shear modulus, activation energy, rutting factor and recovery rate than the SBS-modified asphalt, thus

showing better high-temperature performance. However, the cone penetration and ductility of the GQDs/SBS composite-modified asphalt decrease while S/m increases, indicating that its low-temperature performance is worsened.

(3) The pavement performance of the GQDs/SBS composite-modified asphalt mixture and SBS-modified asphalt mixture are compared. The high-temperature stability of the GQDs/SBS composite-modified asphalt mixture is improved to some extent compared to that of the SBS-modified asphalt mixture, while its water stability changes slightly and the low-temperature performance declines to some extent.

**Author Contributions:** Conceptualization, M.W.; methodology, X.W.; validation, Q.X.; data curation, L.Y.; writing—original draft preparation, Y.L. and N.S.; writing—review and editing, P.Z.; supervision, P.Z.; project administration, P.Z.; funding acquisition, P.Z. All authors have read and agreed to the published version of the manuscript.

**Funding:** This study was monetarily bolstered by the Natural Science Fund project in Shandong province: ZR2021ME189, Project of science and technology support for youth entrepreneurship in Colleges and universities of Shandong Province (2019KJG004).

**Institutional Review Board Statement:** Not applicable.

**Informed Consent Statement:** Not applicable.

**Data Availability Statement:** The information used to bolster the discoveries of this research are from prior detailed researches cited before. This original copy does not include distributed Figures, Tables, and Charts before; thus, all Figures, Tables and Charts of this original copy are unique.

**Acknowledgments:** We would like to recognize numerous co-workers, students and research facility associates for offering specialized assistance on instrument examination.

**Conflicts of Interest:** The authors have no conflict of interest.

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
