# Peer review of "Research on the Preparation of Graphene Quantum Dots/SBS Composite-Modified Asphalt and Its Application Performance"

_coatings, doi:10.3390/coatings12040515_

Round 1

Reviewer 1 Report

Review of the Manuscript ID: coatings-1640921:“Research on Preparation of Graphene Quantum Dots / SBS 2 Composite Modified Asphalt and Its Application Performances” submitted for publication in the journal Coatings.

I have read the manuscript with a great interest because the modifications of asphalt pavement directed to road asphalt improvement is an area that allows both the use of lower value material and achieve improvement in the road asphalt performance. The paper is well designed, structured, and written and presents valuable information for the scientists and engineers working in field of road coatings. I suppose that this paper will receive high citations because the information presented is comprehensive, topical, and original. I am strongly in favor of publication of this article in Coatings journal after minor amendments taking into account the comments made below:

  1. The authors are advised in the sentence (lines 74-78) “Specifically, there are abundant products for nano- modification of binding materials (referring to asphalt and concrete) [11], including the nano-materials of graphene family (e.g. original graphene, monolayer graphene, multi-layer graphene and graphene nanosheets), graphene oxide (GO), single-wall carbon nanotube (CNT) and multi-wall CNT.” to cite the references: Stratiev, D.; Shishkova, I.; Dinkov, R.; Kirilov, K.; Yordanov, D.; Nikolova, R.; Veli, A.; Tavlieva, M.; Vasilev, S.; R. Suyunov. Variation of oxidation reactivity of straight run and H-Oil hydrocracked vacuum residual oils in the process of road asphalt production, Road Mat. Pav. Des. 2021, DOI:10.1080/14680629.2021.1893209. ;

Stratiev, D.; Nenov, S.; Nedanovski, D.; Shishkova, I.; Dinkov, R.; Stratiev, D.D.; Stratiev, D.D.; Sotirov, S.; Sotirova, E.; Atanassova, V.; et al. Empirical Modeling of Viscosities and Softening Points of Straight-Run Vacuum Residues from Different Origins and of Hydrocracked Unconverted Vacuum Residues Obtained in Different Conversions. Energies 2022, 15, 1755. https://doi.org/10.3390/en15051755

They could be cited under references Nrs.11 and 12 in the revised manuscript.

  1. Line 123: “The deoiling asphalt (DOA, asphalt content is 20%)…”. Probably the authors mean asphaltene content of 20%. Please, specify which is the based asphalt used to prepare the deoiled asphalt. The data in Table 1 shows that Qinhuangdao AH-70 asphalt contains 14 wt.% asphaltenes.

Author Response

Comment 1. The authors are advised in the sentence (lines 74-78) “Specifically, there are abundant products for nano- modification of binding materials (referring to asphalt and concrete) [11], including the nano-materials of graphene family (e.g. original graphene, monolayer graphene, multi-layer graphene and graphene nanosheets), graphene oxide (GO), single-wall carbon nanotube (CNT) and multi-wall CNT.” to cite the references: Stratiev, D.; Shishkova, I.; Dinkov, R.; Kirilov, K.; Yordanov, D.; Nikolova, R.; Veli, A.; Tavlieva, M.; Vasilev, S.; R. Suyunov. Variation of oxidation reactivity of straight run and H-Oil hydrocracked vacuum residual oils in the process of road asphalt production, Road Mat. Pav. Des. 2021, DOI:10.1080/14680629.2021.1893209. ;

Stratiev, D.; Nenov, S.; Nedanovski, D.; Shishkova, I.; Dinkov, R.; Stratiev, D.D.; Stratiev, D.D.; Sotirov, S.; Sotirova, E.; Atanassova, V.; et al. Empirical Modeling of Viscosities and Softening Points of Straight-Run Vacuum Residues from Different Origins and of Hydrocracked Unconverted Vacuum Residues Obtained in Different Conversions. Energies 2022, 15, 1755. https://doi.org/10.3390/en15051755

They could be cited under references Nrs.11 and 12 in the revised manuscript.

Response: Thanks for this comment. The two references you mentioned have been cited in the revised manuscript.

Comment 2. Line 123: “The deoiling asphalt (DOA, asphalt content is 20%)…”. Probably the authors mean asphaltene content of 20%. Please, specify which is the based asphalt used to prepare the deoiled asphalt. The data in Table 1 shows that Qinhuangdao AH-70 asphalt contains 14 wt.% asphaltenes.

Response: Thanks for this comment. This section has been clarified in the revised manuscript and is described as follows:“The deoiling asphalt (DOA, Asphaltenes content is 20%) from SINOPEC Jiujiang company was used as the raw material and GQDs with asphaltene polycyclic aromatic hydrocarbon nucleus were prepared by nitric acid oxidation. ”

Reviewer 2 Report

This paper talks about preparing a graphene quantum dots (GQDs)/ styrene-butadiene segmented copolymer composite (GQDs/SBS) as the asphalt modifier by using the Pickering emulsion polymerization method. GQDs/SBS has advantages of simple preparation technique, low cost and being environmental-friendly. It has become a beneficial choice as asphalt cementing material modifier.

Dear authors, 

Very interesting topic, and good work in general.

Please pay attention to grammar formatting, and there are a lot of spaces missing between citations and words and words in general.

Please see below some of the examples. 

Abstract: This aims, it is better to say this study or research aims…

Throughout the entire manuscript ( e.i. L 53, 56..)  – Please fix the spacing between a word and a citation number.

L 74: Attention is plural without s

Space between the bottom of Table 1 and text

L 137: 30 min space

L159 – 7 mg space

L 181:  25 mm space

L189: 1.5mm(8mm plate) space

L 191: l0 rad/s … please check the spacing throughout the entire manuscript, there are a lot of mistakes.

Figure 5: Increase the font for blue and purple text, it is too small to read

Table 3: separate the bottom form the text

Table 4: separate the bottom form the text

Author Response

Reviewer #2: 

This paper talks about preparing a graphene quantum dots (GQDs)/ styrene-butadiene segmented copolymer composite (GQDs/SBS) as the asphalt modifier by using the Pickering emulsion polymerization method. GQDs/SBS has advantages of simple preparation technique, low cost and being environmental-friendly. It has become a beneficial choice as asphalt cementing material modifier.

Dear authors, 

Very interesting topic, and good work in general.

Please pay attention to grammar formatting, and there are a lot of spaces missing between citations and words and words in general.

Please see below some of the examples. 

Comment 1.Abstract: This aims, it is better to say this study or research aims…

Response: Thanks for this comment. This section has been clarified in the revised manuscript and is described as follows: “This study aims to prepare a graphene quantum dots (GQDs)/ styrene-butadiene segmented copolymer composite (GQDs/SBS) as the asphalt modifier by using the Pickering emulsion polymerization method. ”

Comment 2. Throughout the entire manuscript ( e.i. L 53, 56..)  – Please fix the spacing between a word and a citation number.

L 74: Attention is plural without s

Space between the bottom of Table 1 and text

L 137: 30 min space

L159 – 7 mg space

L 181:  25 mm space

L189: 1.5mm(8mm plate) space

L 191: l0 rad/s … please check the spacing throughout the entire manuscript, there are a lot of mistakes.

Figure 5: Increase the font for blue and purple text, it is too small to read

Table 3: separate the bottom form the text

Table 4: separate the bottom form the text

Response: Thanks for this comment. All issues have been corrected in the revised version.

Reviewer 3 Report

The authors wrote a paper: "Research on Preparation of Graphene Quantum Dots / SBS Composite Modified Asphalt and Its Application Performances".
The main goal of this manuscript is preparation and investigation of physicochemical properties of graphene quantum dots (GQDs)/ styrene-butadiene segmented copolymer composite (GQDs/SBS) as the asphalt modifier.

It is very unusual to start a first sentence in the Abstract with:
"This aims to..."

Introduction is poorly written and is missing many references. E.g. this statement has no reference:

"Recently, applications of new materials and new technologies have achieved great progresses. In particular, with the continuous development of nanotechnology, adding nanomaterials into asphalt materials, such as applications of nanoclay, nano-silica, nano-ore and nano-metal, occur frequently."
SBS is never defined nor structure is presented.

It is hard to follow introduction, it's too long and should be completely rewritten.

Page 4.
2. Materials and methods
For FT-IR spectral analysis some basic information are missing (e.g. temperature of the sample).
Additional details about IR spectra are needed.

In abstract and in Conclusion there are statements about VDW forces and chemical covalent bond. There is no discussion on VDW forces or any proof of existence. On page 6 authors state:
"Meanwhile, there are obvious shoulder peaks at 1650cm-1-1580cm-1, which are bending vibration peaks of N-H and C=O in amido bonds"
These peaks do not belong to any bending vibrations and this should be clarified.

Author Response

Reviewer #3: 

The authors wrote a paper: "Research on Preparation of Graphene Quantum Dots / SBS Composite Modified Asphalt and Its Application Performances".
The main goal of this manuscript is preparation and investigation of physicochemical properties of graphene quantum dots (GQDs)/ styrene-butadiene segmented copolymer composite (GQDs/SBS) as the asphalt modifier.

 Comment 1. It is very unusual to start a first sentence in the Abstract with:

"This aims to..."

Response: Thanks for this comment. This section has been clarified in the revised manuscript and is described as follows: “This study aims to prepare a graphene quantum dots (GQDs)/ styrene-butadiene segmented copolymer composite (GQDs/SBS) as the asphalt modifier by using the Pickering emulsion polymerization method. ”

 Comment 2. Introduction is poorly written and is missing many references. E.g. this statement has no reference:

"Recently, applications of new materials and new technologies have achieved great progresses. In particular, with the continuous development of nanotechnology, adding nanomaterials into asphalt materials, such as applications of nanoclay, nano-silica, nano-ore and nano-metal, occur frequently."
SBS is never defined nor structure is presented.

It is hard to follow introduction, it's too long and should be completely rewritten.

Response: Thanks for this comment. The introduction section has been rewritten in the revised version.

Comment 3.  Page 4. 2. Materials and methods

For FT-IR spectral analysis some basic information are missing (e.g. temperature of the sample). Additional details about IR spectra are needed.

Response: Thanks for this comment. This section has been rewritten in the revised version:“The functional groups and material structures of SBS modifier and GQDs/SBS modifier were characterized by using Fourier infrared microscopic analysis spectrometer. Meanwhile, their chemical compositions were analyzed. Nicolet IS 5-type infrared spectrometer (Thermo Science, USA) was used in the experiment. All tests were performed at room temperature.The resolution was 4 cm-1,scanning frequency was 32 times/min and the spectral wave number ranged between 4000 cm-1 and 500 cm-1. The samples were prepared by casting a film onto a potassium bromide (KBr) window from a 5% by weight solution in carbon tetrachloride (CCl4).”

Comment 4. In abstract and in Conclusion there are statements about VDW forces and chemical covalent bond. There is no discussion on VDW forces or any proof of existence. On page 6 authors state:

"Meanwhile, there are obvious shoulder peaks at 1650cm-1-1580cm-1, which are bending vibration peaks of N-H and C=O in amido bonds"
These peaks do not belong to any bending vibrations and this should be clarified.

Response: Thanks for this comment. The description of this part has been removed in the revised version and the functional groups corresponding to the absorption peaks at  1650cm-1-1580cm-1have been redescribed.

Round 2

Reviewer 3 Report

No additional comments.